# Poirot at CMCL 2022 Shared Task: Zero Shot Crosslingual Eye-Tracking Data Prediction using Multilingual Transformer Models

**Harshvardhan Srivastava**
Oracle India , Bengaluru
Indian Institute of Technology, Kharagpur
`harshvardhan.srivastava@oracle.com`

## Abstract

Eye tracking data during reading is a useful source of information to understand the cognitive processes that take place during language comprehension processes. Different languages account for different cognitive triggers, however there seems to be some uniform indicators across languages. In this paper, we describe our submission to the CMCL 2022 shared task on predicting human reading patterns for multilingual dataset. Our model uses text representations from transformers and some hand engineered features with a regression layer on top to predict statistical measures of mean and standard deviation for 2 main eye-tracking features. We train an end-to-end model to extract meaningful information from different languages and test our model on two separate datasets. We compare different transformer models and show ablation studies affecting model performance. Our final submission ranked 4th place for SubTask-1 and 1st place for SubTask-2 for the shared task.

## 1 Introduction

Eye tracking provides an accurate millisecond record of where people are looking while reading and is useful for descriptive study of language processing and understanding of the cognitive processing of brain related to reading. Eye movements are many times language-specific because they depend on structure and ordering of words which are language dependent, however some features tend to be stable and universal and can be observed in all languages. Modeling of human reading has been widely explored in psycholinguistics. The ability to accurately predict eye tracking between languages pushes this field forward, facilitating comparisons between models and analysis of their various functions.

In this paper, we compare our eye-tracking prediction results with some simple baselines using token-level features, which we improve upon with our zero shot cross lingual model which we have described in section 3.

## 2 Task Description

### 2.1 Problem Statement

In this section we briefly describe the task at hand which is the challenge of predicting eye-tracking features recorded during sentence processing of multiple languages. This task is more complex as compared to previous editions of the shared task due to changes made compared to the previous edition (Hollenstein et al., 2021); (i) **Multilingual data**: We use an eye movement dataset with sentences from six languages (Chinese, Dutch, English, German, Hindi, Russian) and (ii) **Eye-tracking features**: To take into account the individual differences between readers, the task is not limited to predict the mean eye tracking features across readers, but also the standard deviation of the feature values. The task details can be found in Hollenstein et al. (2022).

We formulate the task as a regression task to predict 2 eye-tracking features and the corresponding standard deviation across readers. The targets are briefly described here: **first fixation duration (FFDAvg)**, the duration of the first fixation on the prevailing word; **standard deviation (FFDStd)** across readers; **total reading time (TRTAvg)**, the sum of all fixation durations on the current word, including regressions; **standard deviation (TRTStd)** across readers.

The shared task is modelled as two related subtasks of increasing complexity :

**Subtask 1**: Predict eye-tracking features for sentences of the 6 provided languages

**Subtask 2**: Predict eye-tracking features for sentences from a new surprise language

### 2.2 Related Work

Multiple deep learning approaches have been explored in the past on cognitive modelling with lin-

| Name | Abbreviation | Language | Subjects | Training Set | | Dev Set | | Test Set | | Source |
|---|---|---|---|---|---|---|---|---|---|---|
| | | | | Sentences | Tokens | Sentences | Tokens | Sentences | Tokens | |
| Beijing Sentence Corpus | BSC | ZH | 60 | 120 | 1355 | 7 | 82 | 23 | 248 | Pan et al. (2021) |
| Postdam-Allahabad Hindi Eye-tracking Corpus | PAHEC | HI | 30 | 122 | 2021 | 7 | 142 | 24 | 433 | Husain et al. (2015) |
| Russian Sentence Corpus | RSC | RU | 84 | 115 | 1140 | 7 | 59 | 22 | 218 | Laurinavichyute et al. (2019) |
| Provo Corpus | Provo | EN | 30 | 107 | 2067 | 6 | 152 | 21 | 440 | Luke and Christianson (2018) |
| ZuCo 1.0 Corpus (NR) | ZuCo1 | EN | 12 | 240 | 5235 | 15 | 269 | 45 | 994 | Hollenstein et al. (2018) |
| ZuCo 2.0 Corpus (NR) | ZuCo2 | EN | 18 | 279 | 5398 | 17 | 303 | 53 | 1127 | Hollenstein et al. (2019) |
| GECO Corpus (Dutch L1 part) | GECO-NL | NL | 18 | 640 | 7462 | 40 | 405 | 120 | 1475 | Cop et al. (2017) |
| Potsdam Textbook Corpus | PoTeC | DE | 75 | 80 | 1463 | 5 | 139 | 16 | 293 | Jäger et al. (2021) |
| Copenhagen Corpus | CopCo | DA | - | - | - | - | - | 402 | 6767 | - |

Table 1: Overview of the selected datasets

guistic perspective on English datasets ZuCo (Hollenstein et al., 2018, 2019) and Provo(Luke and Christianson, 2018). Salicchi and Lenci (2021) uses cosine similarity and surprisal within regression architecture to model the surprisal characteristic of a new word. Li and Rudzicz (2021); Yu et al. (2021) use the transformer methods to extract the linguistic embeddings; the former applying ensembling methods while the latter using surface, linguistic and behavioral features in combination with the linguistic embeddings.

## 2.3 Dataset

The dataset comprises of the eye-tracking data recorded during natural reading from 8 datasets in 6 languages. The training data contains 1703 sentences, the development set contains 104 sentences, and the test set 324 sentences. The data provided contains scaled features in the range between 0 and 100 to facilitate evaluation via the mean absolute average (MAE). The eye-tracking feature values are averaged over all readers.

The detailed dataset information about the number of sentences in each datasource and the token-wise information is shown in table 1.

## 3 Our Approach

Our models heavily use contextualised embeddings extracted from the pretrained models based on transformer architecture (Vaswani et al., 2017). We experiment on the training dataset with multilingual transformer models which are briefly described below :

**mBERT** (Devlin et al., 2019) a deep contextual representation based on a series of transformers trained by a self-supervised objective with data from Wikipedia in 104 languages. It has been trained with masked language modelling objective

and training makes no use of explicit cross-lingual signal.

**XLM** (Lample and Conneau, 2019) is a Transformer-based model that, like BERT, is trained with the masked language modeling (MLM) objective. Additionally, XLM is trained with a Translation Language Modeling (TLM) objective in an attempt to force the model to learn similar representations for different languages.

**XLM-RoBERTa** (Conneau et al., 2020) uses self-supervised training techniques to achieve state-of-the-art performance in cross-lingual understanding. It is trained on unlabeled text in 100 languages extracted from CommonCrawl datasets.

These transformer methods use either WordPiece or BytePair model for tokenization, due to which we use only the first token embedding of the tokenized word by these methods. We use the above three tranformer models and attach the extracted output embeddings from these models to the manually constructed features which we have described in 3.1. The entire model architecture is explained in Figure 1.

## 3.1 Features

Along with the encoder representations from the multilingual transformer models, we use 3 additional features, which we use to help us provide information to our embeddings. We discard other features like POS-Tag and word_freq, due to non-uniformity in cross-lingual setting and unavailability of reliable and enormous word-frequency list for some of the languages which could reduce the performance and create bias for some languages during training time.

The first two length based features use word division information and the word length information. During neurological processing of language, brain takes up dual pathways to process a word as shown

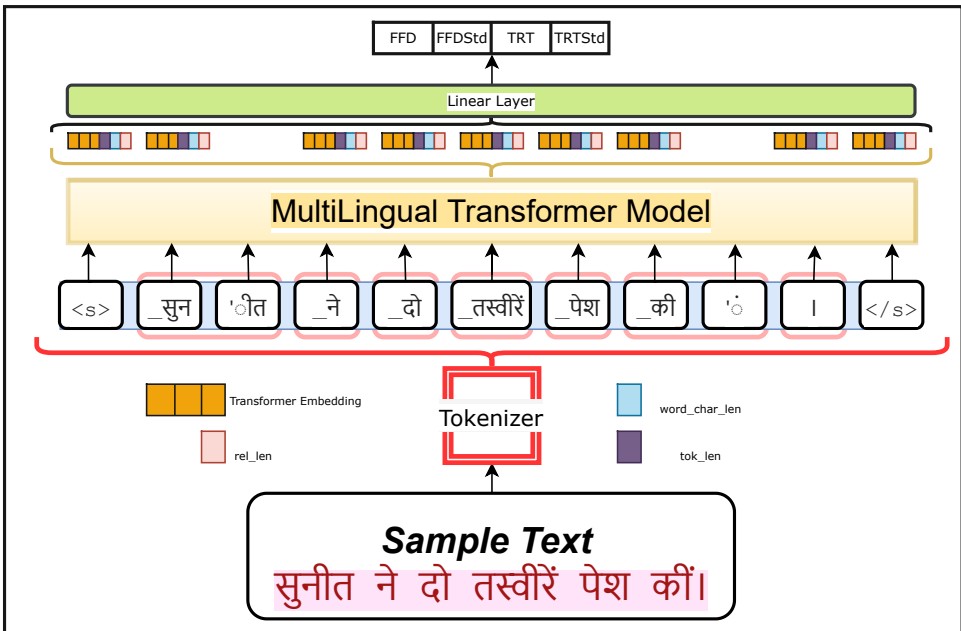

Figure 1: Model Representation

in MacGregor and Shtyrov (2013). The third feature is based on the relative length of the word as compared to preceding word.

**tok_len** : This feature is length of the parts of words when the word is tokenized measured in number of parts, which focuses on the complexity of the word based on length as cognitively longer words as processed.

**word_char_len** : This second feature is based on the apparent space taken up by the word evaluated by the number of characters it takes up when represented in `UTF-8` format. This feature is inspired by studies shown in Joseph et al. (2009).

**rel_len**: The third feature is the relative length of the word as compared to its preceding word. This can capture a sense of ease in reading a short word immediately after reading a word with great character length. For starting words of the sentence, we take the previous word length as zero.

### 3.2 Baselines

We start by implementing some simple baselines using token-level features using length based ideas as word-length is a commonality which can be found in multilingual settings which we described in section 3.1. We start with a median baseline model which takes the median of all the training token labels, instead of using an average baseline to prevent offsetting the predictions by a language with higher or lower valued variables. Along with the median baseline , we also use 5 commonly used machine

learning regression models (i) Linear Regression (*lr*) ,(ii) Support Vector Regression (*svr*),(iii) Gradient Boosting Methods (*lgbm*, *xgboost*) , (iv) Multi-Layer Perceptron (*MLPRegressor*) as our baselines. These baselines do not contain any contextual information.

### 3.3 Implementation and Hyperparameter Details

The models were trained with *MSE*(mean squared error) as loss function and the final evaluation was done using *MAE*(Mean Absolute Error) measure. For the baseline, models were trained taking each label as a regression target variable to remove label correlation if a regressor performs poorly in a multi output regression setting, while when using transformer based models, we trained a single model with a 4 length output regressor head which corresponded each to the final output target variables. Before the final regression layer, we used a hidden linear layer after the embedding output. The model evaluation for task submission was done using dev set after every epoch to measure the performance improvement and prevent overfitting. The implementation can be found here [1].

The details of the hyperparameters used for the training are given in table 4.

---

[1] https://github.com/hvarS/CMCL-2022

| Model | Dev Set | | | | | Test Set - SubTask1 | | | | | Test Set - SubTask2 | | | | |
|---|---|---|---|---|---|---|---|---|---|---|---|---|---|---|---|
| | FFDAvg | FFDStd | TRTAvg | TRTStd | Overall | FFDAvg | FFDStd | TRTAvg | TRTStd | Overall | FFDAvg | FFDStd | TRTAvg | TRTStd | Overall |
| Baseline$_{median}$ | 5.931 | 2.578 | 8.999 | 5.886 | 5.848 | 5.448 | 2.440 | 8.361 | 5.661 | 5.478 | 3.459 | 2.436 | 6.524 | 5.857 | 4.569 |
| Baseline$_{lr}$ | 5.615 | 2.570 | 8.574 | 5.768 | 5.632 | 5.243 | 2.465 | 8.289 | 5.750 | 5.437 | 4.755 | 3.002 | 8.721 | 7.252 | 5.932 |
| Baseline$_{svr}$ | 5.203 | 2.492 | 8.118 | 5.650 | 5.366 | 4.848 | 2.356 | 7.700 | 5.465 | 5.092 | 3.580 | 2.399 | 6.588 | 5.798 | 4.591 |
| Baseline$_{lgbm}$ | 5.209 | 2.528 | 8.004 | 5.534 | 5.319 | 4.835 | 2.415 | 7.869 | 5.584 | 5.176 | 4.390 | 2.966 | 8.407 | 7.127 | 5.723 |
| Baseline$_{MLPRegressor}$ | 5.268 | 2.531 | 8.195 | 5.701 | 5.423 | 4.914 | 2.418 | 7.972 | 5.734 | 5.260 | 4.315 | 2.904 | 8.136 | 7.328 | 5.671 |
| Baseline$_{xgboost}$ | 5.210 | 2.532 | 8.050 | 5.566 | 5.340 | 4.834 | 2.413 | 7.871 | 5.591 | 5.178 | 4.302 | 2.942 | 8.337 | 7.106 | 5.672 |
| mBERT$_{uncased}$ | 5.014 | 2.512 | 7.981 | 5.523 | 5.257 | 4.795 | **2.325** | **7.267** | 5.409 | 4.949 | 3.756 | 3.012 | 5.578 | 5.841 | 4.546 |
| mBERT$_{cased}$ | 5.025 | 2.492 | 8.011 | 5.498 | 5.369 | 4.801 | 2.413 | 7.342 | **5.124** | 4.920 | 3.754 | 3.056 | 5.579 | 5.764 | 4.538 |
| XLM$_{100}$ | 4.914 | 2.584 | 8.134 | 5.512 | 5.286 | 4.902 | 2.425 | 7.814 | 5.414 | 5.138 | 3.331 | 2.944 | 5.448 | 5.798 | 4.380 |
| XLM-RoBERTa$_{base}$ | 4.892 | 2.486 | 8.231 | 5.504 | 5.278 | 4.745 | 2.327 | 7.321 | 5.738 | 5.031 | 3.214 | 2.987 | 5.556 | 5.666 | 4.355 |
| **XLM-RoBERTa$_{large}$** | 4.845 | 2.482 | 7.943 | 5.491 | 5.215 | **4.738** | 2.364 | 7.268 | 5.223 | **4.898** | **2.945** | **2.726** | **5.602** | **5.654** | **4.232** |

Table 2: MAE results on the dev and test set. **Bold** entries are the best performing models for that particular target

| Model Version | SubTask-1 | | | | SubTask-2 | | | |
|---|---|---|---|---|---|---|---|---|
| | FFDAvg | FFDStd | TRTAvg | TRTStd | FFDAvg | FFDStd | TRTAvg | TRTStd |
| XLM-RoBERTa$_{large}$ | 4.738 | 2.364 | 7.268 | 5.223 | 2.945 | 2.726 | 5.602 | 5.654 |
| - tok_len | 4.746 | 2.486 | 7.314 | 5.463 | 2.944 | 2.692 | 5.605 | 5.640 |
| - word_char_len | 4.976 | 2.484 | 7.454 | 5.478 | 3.014 | 2.696 | 5.712 | 5.642 |
| - rel_len | 4.787 | 2.486 | 7.457 | 5.466 | 3.121 | 2.703 | 6.241 | 5.644 |
| - tok_len,word_char_len | 5.012 | 2.427 | 7.785 | 5.421 | 3.154 | 2.710 | 6.785 | 5.546 |
| - tok_len,rel_len | 5.097 | 2.497 | 7.854 | 5.601 | 3.564 | 2.731 | 6.645 | 5.645 |
| - word_char_len,rel_len | 5.124 | 2.492 | 7.771 | 5.671 | 3.452 | 2.722 | 6.621 | 5.664 |
| - tok_len,word_char_len,rel_len | 5.465 | 2.488 | 8.370 | 5.684 | 4.371 | 2.744 | 7.186 | 5.678 |

Table 3: Feature Importance Ablation Study. The best performing model XLM-RoBERTa is taken for ablation

| Parameter | Value |
|---|---|
| Optimizer | AdamW |
| Warm-Up Steps (%) | 10% |
| epochs | 100 |
| learning rate | 5e-2 |
| weight decay | 1e-2 |
| dropout | 0.5 |
| batch size | 64 |
| hidden layer size | 1024 |

Table 4: Hyperparameter Details

# 4 Results and Discussion

Table 2 shows the evaluation results on the dev set and the two test sets for SubTask-1 and SubTask-2 respectively based on MAE on the 4 target variables. Our transformer based models strongly outperformed the baseline approaches. The best performing model was XLM-RoBERTa$_{large}$ model, edging over the transformer models. mBERT model performed better than the XLM model on SubTask-1, but XLM outperforms the former on SubTask-2, suggesting better zero shot performance of XLM for this subtask. Also, the large models tended to perform better than their base counterpart implying higher parameter count resulted in better cross-lingual and zero shot cross-lingual performance. Also since originally the XLM-RoBERTa, mBERT and XLM models were

trained for masked language modelling purpose, they have inherent inner representations of over 100 languages which helps in cross-lingual downstream tasks. One possible reason that mBERT performs better than XLM on SubTask-1 could be that XLM models are used for general sentence representations which mBERT identifies language from context and infers accordingly. For the same mentioned reason, it is possible that XLM performs better in zero shot setting.

## 4.1 Feature Importance

To evaluate the effectiveness of the engineered features ; tok_len, word_char_len and rel_len, an ablation study was conducted using the best performing model. We employ the strategy similar to used in Oh (2021); the three input features were ablated by simply replacing them with zeros during inference, which allowed us to effectively analyse the influence of these additional features. Table 3 shows the effects of model performance without the permutation of the engineered features.

Ablations on the external features show that these features affect the mean ($\mu$) feature values, specifically the *FFDAvg* and the *TRTAvg*, indicating that these external features influence the final model performance for target mean values while the contextualised embedding portion takes care of the standard deviation ($\sigma$) of the targets. It can be observed that the word_char_len feature af-

fects the target FFDAvg value to a large extent, while `rel_len` clearly affects the model performance on SubTask-2. One of the possible reasons could be the infusion of previous word contextual knowledge captured by `rel_len`. Also, the feature `tok_len` in combination with other features also improves the model performance, which may indicate it being not a very strong sole indicator.

## 5 Conclusion and Future Work

In this paper, we presented our approach to the CMCL 2022 Shared Task on eye-tracking data prediction. Our models use the fusion model that involve using the multilingual contextualized token representations using transformer architecture and attaching input features that we created that aid the model performance in predicting eye tracking features. This approach helped us become language agnostic which essentially helped the model to perform well in the zero shot cross-lingual setting in Subtask-2. Our best model based on XLM-RoBERTa outperforms the baseline and is also competitive with other systems submitted to the shared for both SubTask-1 and SubTask-2. Although the embeddings from large language models as shown previously work fairly well as they consider the context of the sentence into consideration as well, possibly they can be improved further if take into consideration the surprisal index which would positively correlate with the reading time and fixation duration as shown in Salicchi and Lenci (2021). In future, we aim to use more etymological features based on shared language history and also use the cross language lexical similarity index when predicting in cross lingual setting .

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
