# OpenReview forum: "Poirot at CMCL 2022 Shared Task: Zero Shot Crosslingual Eye-Tracking Data Prediction using Multilingual Transformer Models"
_aclweb.org/ACL/2022/Workshop/CMCL_Shared_Task — CMCL Shared Task_

### Official Review · Reviewer_9pP4 · 2022-03-21
**A new SOTA paper for crosslingual eye-tracking prediction**

**Rating:** 7
**Confidence:** 4

**Review:**

The paper describes a method for predicting human reading behavior across multiple datasets that can be used for both multilingual and crosslingual setting. The proposed method extracts neural-based embeddings from large language models such as mBERT, XLM, and XLM-RoBERTa and combines them with traditional predictors such as token and sentence-based lengths. This hybrid method is shown to successfully work for the crosslingual prediction setting after obtaining 1st place against other proposed methods. This merits a clear accept for the paper.

Minor questions for improvement of discussion:
1. Figure 1 looks oddly large when the paper is read. I suggest resizing the figure to add more space for supporting discussions and analysis.
2. Although using Transformer-based embeddings produced high results, readers working on the same topic, especially non-CS people, might appreciate insights from the author/s on possible weaknesses of embeddings from large language models compared to theoretically-grounded predictors such as surprisal.
3. Aside from the size or parameter count, what other properties of XLM-RoBERTa do you think made it useful for crosslingual prediction? Based on the results, what property or nature of mBERT made it perform better than XLM for multilingual prediction?

---

### Official Review · Reviewer_3TrC · 2022-03-24
**Combining transformer representations with handcrafted features for the prediction of multi- and cross-lingual eye-tracking reading data**

**Rating:** 7
**Confidence:** 5

**Review:**

This paper presents an approach that utilizes contextualized representations from transformer-based multilingual pretrained models along with 3 length-related handcrafted features for the prediction of multi- and cross-lingual human reading behavior. The results show that the models outperform the mean baselines and attain the 4th and the 1st places in the CMCL 2022 Subtask 1 and Subtask 2, respectively.

**Pros**

The author explores a nice selection of models, involving mBERT, XLM and XLM-RoBERTa.

The author also implements various baselines (median baseline and regression models taking only the handcrafted features).

Extensive results are reported regarding the prediction of each eye-tracking feature per model and feature ablation is conducted on the handcrafted features, which provides insightful information.


**Cons**

I think the model setups should be detailed further. For instance, why did the author use single-length output in baselines and 4-length output in the transformed-based models and would that have an effect on the final results? Why did the author choose to calculate a median baseline and what is the rationale behind the set of selected regression models?

It would be better if you could point to related work using transformers and handcrafted features for similar tasks earlier. For instance, Oh (2021) is mentioned, but only in Section 4.

**Questions**

What happens to rel_len if there is no preceding word?

Did the author notice interesting outcomes depending on the input language?


**Writing**

What is meant by 'brain triggers'? Maybe using a different term here would be better.

In several lines, there seems to be an extra space before the comma (i.e. 'In this paper ,').

end to end -> end-to-end

seperate -> separate

breifly -> briefly

In the XLM paragraph, 'for different languages'

In the XLM-RoBERTa paragraph, add a space before 'It'

'The entire model architecture is explained in Figure 1.' instead of 'in 1'.

3.1 Features: Maybe first mention the features that you used and then, indicate the ones you discarded. It would also be nice if the author can clarify the connection to the dual pathways mentioning compound words.

In 4.1 'one of the possible reasons could be'

In the references, for the CMCL 2022 paper, also write '2022.' after the authors' names

---

### Decision · Program_Chairs · 2022-03-28

Accept